# Synthesis of Novel *N*-Heterocyclic Compounds Containing 1,2,3-Triazole Ring System via Domino, “Click” and RDA Reactions

**DOI:** 10.3390/molecules24040772

**Published:** 2019-02-21

**Authors:** Márta Palkó, Mohamed El Haimer, Zsanett Kormányos, Ferenc Fülöp

**Affiliations:** 1Institute of Pharmaceutical Chemistry, University of Szeged, Interdisciplinary excellence centre, Eötvös utca 6, Szeged H-6720, Hungary; palko@pharm.u-szeged.hu (M.P.); el.haimer.mohamed@pharm.u-szeged.hu (M.E.H.); zsani0424@gmail.com (Z.K.); 2MTA-SZTE Stereochemistry Research Group, Hungarian Academy of Sciences, Eötvös utca 6, Szeged H-6720, Hungary

**Keywords:** domino ring closure, click reaction, RDA reaction, stereoselectivity, regioselectivity, traceless chirality transfer

## Abstract

An uncomplicated, high-yielding synthetic route has been developed to constitute complicated heterocycles, applying domino, click and retro-Diels–Alder (RDA) reaction sequences. Starting from 2-aminocarboxamides, a new set of isoindolo[2,1-*a*]quinazolinones was synthesized with domino ring closure. A click reaction was performed to create the 1,2,3-triazole heterocyclic ring, followed by an RDA reaction resulting in dihydropyrimido[2,1-*a*]isoindole-2,6-diones. The absolute configuration, concluded by the norbornene structure that served as a chiral source, remained constant throughout the transformations. The structure of the synthesized compounds was examined by ^1^H and ^13^C Nuclear Magnetic Resonance (NMR) methods.

## 1. Introduction

Compounds containing quinazoline and triazole heterocyclic rings play a significant role in both organic and pharmaceutical chemistries. Quinazoline heterocycles are important subunits of a broad diversity of synthetic pharmaceuticals as well as natural compounds with antiviral [1,2], anti-inflammatory [3], antimalarial [4], and anticancer [5] activities. The synthesis of bioactive saturated quinazoline derivatives was also investigated [6,7,8]. In the past few years, researchers have exploited heterocycles containing the 1,2,3-triazole ring to generate many medicinal scaffolds exhibiting anti-HIV [9], antibacterial [10,11,12] and anticancer activities [13,14]. Building a triazole ring into a compound can change or even improve the pharmacokinetic properties of a drug [15,16,17,18,19,20,21,22,23].

There exist numerous methods, which are effective without using the classic and time-consuming protection–deprotection processes as well as purification methodology of intermediates [24,25,26,27] in the synthesis of diverse and complex chiral compounds from simple substrates in an economically suitable manner. At the same time, the use of multi-component domino reactions in asymmetric synthesis has gained a continuously increasing interest [28,29,30,31,32]. Asymmetric domino reactions are based on the application of removable chiral auxiliaries and chiral reagents [24].

The 1,4-disubstituted 1,2,3-triazole function still plays an important role in drug discovery, which justifies the continuous advancement of new strategies for their synthesis [33,34,35]. One of the most widely used click-chemistry methods in this field is the copper-catalyzed 1,3-dipolar cycloaddition between an alkyne and an azide (CuAAC), due to its simplicity and high selectivity [36,37,38,39].

For synthetic chemists, the RDA reaction has become a valuable tool for their research towards the design and synthesis of novel heterocyclic scaffolds, because of its efficiency in introducing a double bond into a heterocyclic skeleton [40] along with enantiocontrolled [41] and enantiodivergent [42] syntheses of heterocyclic compounds.

The skeletal transformations of heterocycles take its place in the construction of complex molecular frameworks from simple feedstock among the most powerful synthetic strategies [43,44,45]. Within this context, domino ring closure and RDA are the predominant approaches, since they lead to valuable nitrogenated heterocycles of high biological activity [46,47,48,49,50,51,52,53], such as isoindolo- and pyrroloquinazolinones. In our laboratory, their reactivity and skeletal deformation have been widely examined under mild conditions.

Continuing our work on the synthesis of novel *N*-heterocycles and focusing on the biological potential of fused quinazolinones [47,48,49,50,51,52,53], herein we report the synthesis of a new series of isoindolo[2,1-*a*]quinazolinones and pyrimido[2,1-*a*]isoindoles starting from 2-aminocarboxamides.

Our present aim was (*i*) to examine the diastereoselectivity of the domino ring-closure reaction of *N*-propargyl-substituted *diendo*- and *diexo*-2-aminonorbornenecarboxamides with 2-formylbenzoic acid, (*ii*) to develop Cu(I)-catalyzed azide/alkyne cycloaddition (CuAAC) in a regioselective manner, (*iii*) to investigate the RDA reaction of the created isoindolo[2,1-*a*]quinazolinones, and (*iv*) to extend this methodology to obtain different racemic and enantiomeric pyrimido[2,1-*a*]isoindole derivatives containing a triazole ring.

## 2. Results

First, all methods were performed and optimized with racemic starting materials followed by repeating the syntheses with the enantiomers. In Scheme 1 and Scheme 2 only a single enantiomer is represented for evidence. Please note that all spectroscopic data of the racemic compounds were identical to those of the enantiomeric samples.

Racemic *N*-Boc-protected amino acids (±)-**3** and (±)-**12** were produced from the corresponding *diendo*- and *diexo*-3-aminobicyclo[2.2.1]hept-5-ene-2-carboxylic acids (±)-**2** and (±)-**11** according to an earlier procedure [54]. The synthesis of enantiomeric Boc-protected amino acids (−)-**3** and (−)-**12** started from enantiomeric 2-aminonorbornene esters (−)-**1** and (−)-**10**, as depicted in Scheme 1 and Scheme 2. The starting enantiomeric 2-aminonorbornene esters (−)-**1** and (−)-**10** were prepared from racemic amino esters (±)-**1** and (±)-**4** by diastereomeric salt formation as previously described [45].

The free enantiomeric amino esters (−)-**1** (*ee* > 95%) and (−)-**10** (*ee* > 96%) were hydrolyzed with water under microwave irradiation to furnish amino acids (−)-**2** and (–)-**11**, which were then reacted with *di-tert-butyl dicarbonate* affording *N*-Boc-protected amino acids (−)-**3** and (−)-**12**. They were then transformed into propargylamides (−)-**4** and (−)-**13** in tetrahydrofuran using propargylamine in the presence of *N,N*′-diisopropylcarbodiimide (DIC) and hydroxybenzotriazole (HOBt). After acidic deprotection of amides (−)-**4** and (−)-**13**, free amide bases (+)-**5** and (+)-**14** were used in the next step without purification.

The reaction of propargylamides **5** and **14** with 2-formylbenzoic acid is plausibly interpreted as a domino process, in which the first step is the Schiff base formation [47,50]. The Schiff base undergoes ring closure to give isoindolo[2,1-*a*]quinazolinones **6** and **15,** respectively.

The reaction was implemented by dissolving (+)-**5** and (+)-**14** in ethanol using one equivalent of 2-formylbenzoic acid and stirring the solution at 100 °C for 30 min under microwave irradiation in the presence of *p*-toluenesulfonic acid. The ^1^H NMR spectra revealed the formation of the isoindolo[2,1-*a*]quinazoline (−)-**6** and (−)-**15** (Scheme 1 and Scheme 2).

The full NMR signal assignment was carried out for compounds (−)-**6** and (−)-**15**. The relative configuration of the new hydrogen in the product from the *diexo* isomer, according to the characteristic NOE crosspeaks, is in *trans* arrangement with the annelated hydrogen atoms in (–)-**6**. The characteristic structure crosspeak was found for (−)-**6** between protons C(6a)-H and C(13)-H. For the *diendo* isomer, in the product of the ring closure reaction, the annelated hydrogen atoms and the relative configuration of the new hydrogen are *trans* in (−)-**15**. For (−)-**15** the characteristic structure crosspeaks were found between the C(6a)-H and C(3)-H protons.

In the next step the 1,2,3-triazole ring was formed by click reaction using Cu(I)-catalyzed azide/alkyne (CuAAc) cycloaddition. The azide was synthesized in situ by dissolving sodium azide in the mixture of 2-methylbenzyl chloride, trimethylamine, and *tert*-butyl alcohol and stirring at room temperature for 1 h [35]. Afterwards, (−)-**6** or (−)-**15** was added along with copper(II) sulfate and sodium ascorbate as a reducing agent. The nascent copper(I) acting as the catalyst is responsible for the regioselectivity [35]. The CuAAC reaction of the terminal alkyne moiety of (−)-**6** and (−)-**15** was completely regioselective affording 1,4-disubstituted triazoles (−)-**7** and (−)-**16**.

In the last step, RDA reaction was performed with (−)-**6**, (−)-**7** and (−)-**15**, (−)-**16** resulting in (−)-**8**, (−)-**9** and (+)-**8**, (+)-**9**, respectively. The reactions were carried out in 1,2-dichlorobenzene by stirring under microwave irradiation at 220 °C for 60 min.

An alternative pathway for the synthesis of (−)-**9** and (+)-**9** was also tested. It was implemented after the RDA reaction of (−)-**6** and (−)-**15** and, as expected, it provided the product molecules with the same structures formed in the original pathway (Scheme 1 and Scheme 2).

## 3. Materials and Methods 

### 3.1. General Methods

^1^H NMR spectra were recorded at 500.20 MHz, while the ^13^C NMR spectra were measured at 125.62 MHz in CDCl_3_ or in DMSO-d6 at ambient temperature, with a Bruker AV NEO Ascend 500 spectrometer (Bruker Biospin, Karlsruhe, Germany) with Double Resonance Broad Band Probe (BBO). Chemical shifts are given, relative to tetramethysilane (TMS) as internal standard, in δ (ppm). Elemental analyses were performed with a Perkin–Elmer CHNS-2400 Ser II Elemental Analyzer. Microwave-promoted reactions were carried out in sealed reaction vials (10 mL) in a microwave (CEM, Discover, SP) cavity (CEM Corporation, Matthwes, NC, USA). Optical rotations were measured with a Perkin–Elmer 341 polarimeter (Perkin–Elmer, Shelton, CT, USA). Melting points were determined with a Hinotex-X4 micro melting point apparatus (Hinotek, Ningbo, China) and are uncorrected. Racemic *diendo*- and *diexo*-3-aminobicyclo[2.2.1]hept-5-ene-2-carboxylic acids (±)-**2** and (±)-**11** were prepared according to a literature procedure [55,56].

The enantiomers of 2-aminonorbornene esters (+)-**1**, (−)-**1**, (+)-**10** and (−)-**10** were prepared from racemic 2-aminonorbornene esters via diastereomeric salt formation with *O,O*′-di-*p*-toluoyltartaric acid (DPTTA) and *O,O*′-dibenzoyltartaric acid (DBTA) as previously published [45]. The *ee* values of (+)-**1** and (−)-**1** were determined by a literature method [45]. The *ee* values for (+)-**10** (92%) and (−)-**10** (98%) were determined by HPLC using Phenomenex-IA column after derivatization with benzoyl chloride in the presence of TEA. Analytical conditions were as follows: eluent: a mixture of *n*-hexane and isopropyl alcohol (IPA) (70:30), flow rate: 0.3 mL·min^−1^, detection at 254 nm, retention times: (−)-**10**: 40.81 min (antipode: 25.33 min). The *ee* values for (−)-**4** (98%), (+)-**4** (95%), (−)-**13** (95%) and (+)-**13** (96%) were determined by GC on a Chirasil-L-Val column (25 m): 180 °C isotherm, 1 mL·min^−1^, retention times (−)-**13**: 4.91 min, (+)-**13**: 4.52 min; flow rate 1 mL·min^−1^, 160 °C for 5 min → 180 °C (rate of temperature rise 10 °C/min; retention times (−)-**4**: 11.84 (min), (+)-**4**: 12.64 (min). The *ee* values of the domino ring closure compounds (−)-**6** and (+)-**6** were identified by HPLC using Chiracel-OD-H column, eluent: a mixture of *n*-hexane and IPA (70:30), flow rate: 0,15 mL·min^−1^, detection at 254 nm, retention times (−)-**6**: 61.83 min, (+)-**6**: 66.29 min. The *ee* values of the domino ring closure products (−)-**15** and (+)-**15** were identified by HPLC using Phenomenex-IA column, eluent: a mixture of *n*-hexane and IPA (60:40), flow rate: 1 mL·min^−1^, detection at 254 nm, retention times (−)-**15**: 22.04 min, (+)-**15**: 42.99 min. The *ee* values of the RDA products (−)-**8** and (+)-**8** were determined by HPLC using Phenomenex-IA column, eluent: a mixture of *n*-hexane and IPA (70:30 containing 0.1% DEA), flow rate: 0.5 mL·min^−1^, detection at 254 nm, retention times (−)-**8**: 80.88 min, (+)-**8**: 76.58 min. The *ee* values of RDA products (−)-**9** and (+)-**9** were determined by HPLC using ChiralPak-IA column, eluent: a mixture of *n*-hexane and IPA (60:40 containing 0.1% DEA), flow rate: 0.5 mL·min^−1^, detection at 254 nm, retention times (−)-**9**: 26.76 min, (+)-**9**: 23.87 min.

### 3.2. Synthesis of New Compounds

#### 3.2.1. Synthesis of Amino Acids [(+)-**2**, (−)-**2**, (+)-**11**, (−)-**11**] 

2-Aminonorbornene ester hydrochlorides (+)-**1**, (−)-**1** [45] or (+)-**10**, (−)-**10** [45] (3.00 g, 13.82 mmol in 30 mL H_2_O) were treated with 10% aqueous NaOH solution (30 mL) to liberate the free acids. The aqueous layer was extracted with CHCl_3_ (3 × 30 mL). The combined organic phase was dried (Na_2_SO_4_) and the solvent was evaporated. The residue was diluted in water (2 mL) in a 10-mL pressurized reaction vial, and the solution was stirred and warmed to 100 °C for 30 min at max. 300 W microwave irradiation. The solvent was evaporated, and the crude product was filtered off from acetone and recrystallized from water/acetone. [(−)-**2**]: white crystals (90% yield), m.p. 258–262 °C. Lit m.p. >260 °C, ${\left[\alpha \right]}_{D}^{20}$ = −12.1 (c = 0.5, H_2_O), lit [57] = ${\left[\alpha \right]}_{D}^{20}$= −12.3 [57] (c = 0.3, H_2_O). [(+)-**2**]: white crystals (91% yield), m.p. 257–260 °C ${\left[\alpha \right]}_{D}^{20}$ = +12.0 (c = 0.5, H_2_O). Lit. [58] ${\left[\alpha \right]}_{D}^{20}$ = +13.8 (c = 1, H_2_O). [(−)-**11**]: white crystals (85% yield) m.p. 236–238 °C, ${\left[\alpha \right]}_{D}^{20}$ = −62 (c = 0.5, H_2_O). [(+)-**11**]: white crystals (91% yield) m.p. 233–237 °C, ${\left[\alpha \right]}_{D}^{20}$ = +62 (c = 1, H_2_O). ^1^H and ^13^C NMR data of the enantiomeric amino acids were identical with those of the racemic compounds [55,56].

#### 3.2.2. Synthesis of Boc-Protected Amino Acids [(+)-**3**, (−)-**3**, (+)-**12**, (−)-**12**]

To a solution of the appropriate 2-aminonorbornene carboxylic acid [(+)-**2**, (−)-**2** (+)-**11** or (−)-**11**, 3.1 g, 10 mmol] in 100 mL of a 2:1 dioxane/H_2_O mixture, 25 mL 1 M NaOH was added. The solution was cooled to 0 °C and Boc_2_O (2.4 g, 11 mmol) was added. The mixture was stirred at 0 °C for 1 h, warmed to r.t., stirred for 4 h and, finally, it was evaporated to 30 mL. The solvent was acidified with 10% H_2_SO_4_ (pH = 2.5) and the resulting mixture was extracted with EtOAc (3 × 50 mL). The solvent was evaporated, and the crude product was filtered off from Et_2_O and recrystallized from iPr_2_O. [(−)-**3**]: white crystals (87% yield), m.p. 133–135 °C, ${\left[\alpha \right]}_{D}^{20}$ = −78 (c = 1, EtOH). [(+)-**3**]: white crystals (91% yield), m.p. 136–137 °C, ${\left[\alpha \right]}_{D}^{20}$ = +76 (c = 1, EtOH). [(−)-**12**]: white crystals (85% yield), m.p. 132–135 °C, ${\left[\alpha \right]}_{D}^{20}$ = −14 (c = 1, EtOH). [(+)-**12**]: white crystals (91% yield), m.p. 133–134 °C, ${\left[\alpha \right]}_{D}^{20}$ = +13 (c = 1, EtOH). ^1^H and ^13^C NMR data of the enantiomeric amino acids were identical with those of the racemic compounds [54,55,56,57,58,59].

#### 3.2.3. Synthesis of Boc-Protected Propargyl Amides [(+)-**4**, (−)-**4**, (+)-**13**, (−)-**13**] 

A mixture of the appropriate amino acid [(+)-**3**, (−)-**3** (+)-**12** or (−)-**12**, 2.53 g, 10 mmol], hydroxybenzotriazole (1.83 g, 12 mmol), *N,N*′-diisopropylcarbodiimide (DIC) (1.51 g, 12 mmol), and propargylamine (0.55 g, 0.7 mL, 10 mmol) was stirred in THF (50 mL) overnight at r.t. After completion of the reaction (checked by thin layer chromatography), the solvent was evaporated. Purification of the residue by column chromatography over silica gel with EtOAc gave the desired products.

*tert-Butyl ((1R,2R,3S,4S)-3-(prop-2-yn-1-ylcarbamoyl)bicyclo[2.2.1]hept-5-en-2-yl)carbamate* [(−)-**4**]: White crystals (75% yield), m.p. 128–130 °C, ${\left[\alpha \right]}_{D}^{20}$ = −38 (c = 1, EtOH), ^1^H-NMR (500 MHz, CDCl_3_, 30 °C): δ = 1.42 (s, 9H, CH_3_) 1.60 (m, 1H, H-7), 2.10 (m, 1H, H-7), 2.20 (t, *J* = 2.3 Hz, 1H, C≡CH) 2.35 (d, *J* = 8.2 Hz, 1H, H-2), 2.69 (m, 1H, H-1), 2.98 (m, 1H, H-4), 3.85–3.92 (m, H, H-4), 4.11–4.16 (m, 1H, NCH_2_), 5.09–5.21 (m, 1H, NCH_2_) 5,91 (s, 1H, NH) 6.14–6.23 (m, 2H, H-5, H-6) ^13^C NMR (125 MHz, CDCl_3_, 30 °C): δ = 28.4, 29.4, 44.9, 45.6, 48.0, 48.1, 53.0, 71.7, 79.3, 79.5, 137.5, 138.8, 155.9, 172.8 ppm. C_16_H_22_N_2_O_3_ (290.36): calcd. C, 66.18; H, 7.64; N, 9.65; found C 66.44; H 7.91; N 9.42. 

*tert-Butyl ((1S,2S,3R,4R)-3-(prop-2-yn-1-ylcarbamoyl)bicyclo[2.2.1]hept-5-en-2-yl)carbamate* [(+)-**4**]: White crystals (74% yield). m.p. 129–131 °C, ${\left[\alpha \right]}_{D}^{20}$ = +37 (c = 1, EtOH).

*tert-Butyl ((1R,2S,3R,4S)-3-(prop-2-yn-1-ylcarbamoyl)bicyclo[2.2.1]hept-5-en-2-yl)carbamate* [(−)-**13**]: White crystals (72% yield), m.p. 161–163 °C, ${\left[\alpha \right]}_{D}^{20}$ = −3.7 (c = 0.5, EtOH), ^1^H-NMR (500 MHz, CDCl_3_, 30 °C): δ = 1.29–1.50 (m, 11H, CH_3_, H-7, H-7) 2.19 (t, *J* = 2.3 Hz, 1H, C≡CH) 2.98–3.10 (m, 3H, H-1, H-2, H-4), 3.75–3.83 (m, H, H-3), 4.03–4.13 (m, 1H, NCH_2_), 4.78–4.86 (m, 1H, NCH_2_) 6.06–6.11 (m, 1H, H-6) 6.20 (s, 1H, NH) 6.59–6.61 (m, 1H, H-5), ^13^C NMR (125 MHz, CDCl_3_, 30 °C): δ = 28.4, 29.4, 46.3, 47.1, 47.7, 53.0, 71.7, 79.3, 79.5, 137.5, 138.8, 155.9, 172.8 ppm. C_16_H_22_N_2_O_3_ (290.36): calcd. C, 66.18; H, 7.64; N, 9.65; found C 66.41; H 7.71; N 9.52.

*tert-Butyl ((1S,2R,3S,4R)-3-(prop-2-yn-1-ylcarbamoyl)bicycle[2.2.1]hept-5-en-2-yl)carbamate* [(+)-**13**]: White crystals (74% yield), m.p. 160–163 °C, ${\left[\alpha \right]}_{D}^{20}$ = +4.1 (c = 0.5, EtOH).

#### 3.2.4. Synthesis of Domino Ring Closure Products [(−)-**6**, (+)-**6**, (−)-**15** and (+)-**15**]

The mixture of Boc-protected amides (+)-**4**, (−)-**4** or (+)-**13**, (−)-**13** (2.32 g, 8 mmol in 15 mL H_2_O) was stirred with 10% aqueous HCl solution (10 mL) at r.t. for 6 h. The aqueous layer was neutralized with 10% aqueous NaOH solution and extracted with CH_2_Cl_2_ (3 × 30 mL). The combined organic phase was dried (Na_2_SO_4_) and the solvent was evaporated. The resulting amides were used for the next steps without purification. Amide (+)-**5**, (−)-**5**, (+)-**14** or (−)-**14** (1.33 g, 7 mmol), 2-formylbenzoic acid (1.05 g, 7 mmol), and *p*TSA (20 mol%) were dissolved in EtOH (5 mL) in a 10-mL pressurized reaction vial, and the solution was warmed to 100 °C for 30 min at max. 300 W microwave irradiation. Upon completion of the reaction (monitored by TLC) the solvent was evaporated and the crude solid was filtered off from Et_2_O and recrystallized from EtOH.

*(1S,4R,4aR,6aS,12aS)-6-(Prop-2-yn-1-yl)-1,4,4a,6,6a,12a-hexahydro-1,4-methanoisoindolo[2,1-a]quinazoline-5,11-dione* [(−)-**6**]: White crystals (71% yield), m.p. 189–191 °C, ${\left[\alpha \right]}_{D}^{20}$= −19 (c = 0.5, EtOH), ^1^H-NMR (500 MHz, CDCl_3_, 30 °C): δ = 1.66–1.80 (m, 2H, H-13, H-13) 2.45 (t, *J* = 2.4 Hz, 1H, C≡CH), 2.63–2.69 (m, 1H, H-4a) 2.91–2.96 (m, 1H, H-4), 3.46–3.48 (m, 1H, H-1), 3.80–3.83 (m, 1H, H-12a), 4.53–4.59 (m, 1H, NCH_2_), 5.40–5.52 (m, 1H, NCH_2_) 6.18 (s, 1H, H-6a) 6.35–6.44 (m, 2H, H-2, H-3) 7.59–8.10 (m, 4H, Ar), ^13^C NMR (125 MHz, CDCl_3_, 30 °C): δ = 32.5, 42.4, 45.7, 49.2, 50.1, 50.4, 68.1, 74.5, 77.8, 125.1, 125.3, 130.8, 132.5, 138.5, 138.8, 139.5, 167.2, 171.3. ppm. C_19_H_16_N_2_O_2_ (304.34.36): calcd. C, 74.98; H, 5.30; N, 9.20; found C 74.91; H 5.42; N 9.35.

*(1R,4S,4aS,6aR,12aR)-6-(Prop-2-yn-1-yl)-1,4,4a,6,6a,12a-hexahydro-1,4-methanoisoindolo[2,1-a]quinazoline-5,11-dione* [(+)-**6**]: White crystals (69% yield), m.p. 189–192 °C, ${\left[\alpha \right]}_{D}^{20}$ = +18 (c = 0.5, EtOH).

*(1S,4R,4aS,6aR,12aR)-6-(Prop-2-yn-1-yl)-1,4,4a,6,6a,12a-hexahydro-1,4-methanoisoindolo[2,1-a]quinazoline-5,11-dione* [(−)-**15**]: White crystals (70% yield), m.p. 189–191 °C, ${\left[\alpha \right]}_{D}^{20}$ = −34.9 (c = 0.5, EtOH), ^1^H-NMR (500 MHz, CDCl_3_, 30 °C): δ = 1.61–1.62 (m, 2H, H-13, H-13) 2.40 (t, *J* = 2.1 Hz, 1H, C≡CH), 3.10–3.19 (m, 1H, H-4a) 3.25–3.35 (m, 1H, H-4), 3.47–3.54 (m, 1H, H-1), 3.60–3.73 (m, 1H, H-12a), 5.22–5.39 (m, 2H, NCH_2_), 5.97 (s, 1H, H-6a), 6.26–6.31 (m, 1H, H-3), 6.44–6.52 (m, 1H, H-2) 7.52–7.96 (m, 4H, Ar), ^13^C NMR (125 MHz, CDCl_3_, 30 °C): δ = 32.2, 32.5, 42.4, 45.3, 49.2, 50.1, 50.4, 68.1, 74.5, 78.2, 125.1, 155.3, 130.8, 132.5, 138.5, 138.8, 139.5, 167.2, 171.3. ppm. C_19_H_16_N_2_O_2_ (304.34.36): calcd. C, 74.98; H, 5.30; N, 9.20; found C 74.82; H 5.51; N 9.32.

*(1R,4S,4aR,6aS,12aS)-6-(Prop-2-yn-1-yl)-1,4,4a,6,6a,12a-hexahydro-1,4-methanoisoindolo[2,1-a]quinazoline-5,11-dione* [(+)-**15**]: White crystals (74% yield), m.p. 189–192 °C, ${\left[\alpha \right]}_{D}^{20}$ = +36 (c = 0.5, EtOH).

#### 3.2.5. General Procedure for the Synthesis of (−)-**7**, (+)-**7**, (−)-**16** and (+)-**16** by Click Reaction

A solution of 2-methylbenzyl chloride (1.1 mmol), NaN_3_ (1 mmol), and Et_3_N (1.3 mmol) in H_2_O/tBuOH (1:1, 10 mL) was stirred at r.t. for 1 h. Subsequently, a mixture of 1,4-methanoisoindolo[2,1-*a*]quinazoline-5,11-dione derivative ((−)-**6**, (+)-**6**, (−)-**15** or (+)-**15**; 1 mmol), sodium ascorbate (0.15 mmol), and CuSO_4_ (10 mol%) was added to the freshly prepared azide derivative and the mixture was stirred at r.t. for 8 h. The mixture was diluted with H_2_O and extracted with EtOAc. The organic phase was dried (Na_2_SO_4_), the solvent was evaporated off. Purification of the residue by column chromatography over silica gel with EtOAc gave the desired products. (The same procedure was used for the synthesis of (−)-**9** and (+)-**9** by click reaction starting from (−)-**8** and (+)-**8**).

*(1S,4R,4aR,6aS,12aS)-6-((1-(2-Methylbenzyl)-1H-1,2,3-triazol-4-yl)-methyl)-1,4,4a,6,6a,12a-hexahydro-1,4-methanoisoindolo[2,1-a]quinazoline-5,11-dione* [(−)-**7**]: White crystals (70% yield), m.p. 100–107 °C, ${\left[\alpha \right]}_{D}^{20}$ = −20 (c = 0.5, EtOH), ^1^H-NMR (500 MHz, CDCl_3_, 30 °C): δ = 1.31–1.42 (m, 2H, H-13) 2.26 (s, 3H, CH_3_), 2.54–2.58 (m, 1H, H-4a) 2.81–2.85 (m, 1H, H-4), 3.17–3.21 (m, 1H, H-1), 4.50–4.54 (m, 1H, H-12a), 4.57–4.64 (d, *J* = 15.9 Hz, 1H, NCH_2_), 5.35–5.42 (d, *J* = 15.9 Hz, 1H, NCH_2_), 5.53 (s, 2H, NCH_2_Ar) 6.10 (s, 1H, H-6a), 6.26–6.41 (m, 2H, H-2, H-3), 7.03–7.33 (m, 4H, Ar), 7.37 (s, 1H, CHtriazol) 7.52–8.34 (m, 4H, Ar), ^13^C NMR (125 MHz, CDCl_3_, 30 °C): δ = 19.0, 37.5, 41.9, 44.5, 48.6, 49.4, 49.6, 52.4, 68.1, 122.6, 124.4, 126.5, 126.7, 129.2, 129.3, 130.3, 131.0, 131.9, 132.1, 132.4, 136.7, 138.2, 139.6, 144.0, 166.9, 171.5. ppm. C_27_H_25_N_5_O_2_ (451.52): calcd. C, 71.82; H, 5.58; N, 15.51; found C 71.93; H 5.47; N 15.35.

*(1R,4S,4aS,6aR,12aR)-6-((1-(2-Methylbenzyl)-1H-1,2,3-triazol-4-yl)-methyl)-1,4,4a,6,6a,12a-hexahydro-1,4-methanoisoindolo[2,1-a]quinazoline-5,11-dione* [(+)-**7**]: White crystals (74% yield), m.p. 100–102 °C, ${\left[\alpha \right]}_{D}^{20}$ = +20 (c = 0.5, EtOH).

*(1S,4R,4aS,6aR,12aR)-6-((1-(2-Methylbenzyl)-1H-1,2,3-triazol-4-yl)-methyl)-1,4,4a,6,6a,12a-hexahydro-1,4-methanoisoindolo[2,1-a]quinazoline-5,11-dione* [(−)-**16**]: White crystals (72% yield), m.p. 128–130 °C, ${\left[\alpha \right]}_{D}^{20}$ = −20 (c = 0.5, EtOH), ^1^H-NMR (500 MHz, CDCl_3_, 30 °C): δ = 1.53–1.65 (m, 2H, H-13) 2.28 (s, 3H, CH_3_), 3.08 (m, 1H, H-4a) 3.20–3.25 (m, 1H, H-4), 3.28–3.33 (m, 1H, H-1), 4.45–4.55 (m, 1H, H-12a), 5.20–5.29 (m, 2H, NCH_2_), 5.47 (m, 2H, NCH_2_Ar) 5.64 (m, 1H, H-3), 5.79 (s, 1H, H-6a), 5.98 (m, 1H, H-2) 7.03–7.32 (m, 4H, Ar), 7.40 (s, 1H, CHtriazol) 7.51–8.40 (m, 4H, Ar), ^13^C NMR (125 MHz, CDCl_3_, 30 °C): δ = 19.0, 37.2, 42.7, 47.8, 48.8, 48.9, 50.0, 52.4, 68.6, 122.9, 124.2, 126.8, 126.8, 129.3, 129.4, 130.0, 131.1, 131.9, 131.9, 132.4, 135.4, 136.7, 137.0, 139.7, 144.0, 167.5, 171.2 ppm. C_27_H_25_N_5_O_2_ (451.52): calcd. C, 71.82; H, 5.58; N, 15.51; found C 71.91; H 5.43; N 15.32.

*(1R,4S,4aR,6aS,12aS)-6-((1-(2-Methylbenzyl)-1H-1,2,3-triazol-4-yl)-methyl)-1,4,4a,6,6a,12a-hexahydro-1,4-methanoisoindolo[2,1-a]quinazoline-5,11-dione* [(+)-**16**]: White crystals (74% yield), m.p. 97–99 °C, ${\left[\alpha \right]}_{D}^{20}$ = +19.3 (c = 0.5, EtOH).

#### 3.2.6. RDA Protocols for the Synthesis of Pyrimido[2,1-a]isoindols [(−)-**8** (+)-**8**, (−)-**9**, (+)-**9**] 

Isoindoloquinazoline derivatives [(−)-**6**, (+)-**6**, (−)-**7**, (+)-**7**, (−)-**15**, (+)-**15**, (−)-**16**, (+)-**16**, 0.5 mmol] were dissolved in 5 mL 1,2-dichlorobenzene. The solution was stirred and heated to 220 °C for 60 min at max. 300 W microwave irradiation. Then the solvent was evaporated, the residue was dissolved in EtOAc and purified by column chromatography on silica gel eluting with EtOAc.

*(S)-1-(Prop-2-yn-1-yl)-1,10b-dihydropyrimido[2,1-a]isoindole-2,6-dione* [(−)-**8**]: White crystals (52% yield), m.p. 91–93 °C, ${\left[\alpha \right]}_{D}^{20}$ = −376 (c = 0.2, EtOH), ^1^H-NMR (500 MHz, CDCl_3_, 30 °C): δ = 2.45 (t, *J* = 2.5 Hz, 1H, C≡CH), 3.73 (dd, *J* = 17.7, 2.1, 1H,CH_2_) 5.15 (d, *J* = 17.7 Hz, 1H, CH_2_) 5.54 (d, *J* = 7.6 Hz, 1H, H-3) 6.20 (s, 1H, H-10b) 7.66–8.10 (m, 5H, Ar, H-4), ^13^C NMR (125 MHz, CDCl_3_, 30 °C): δ = 31.3, 69.8, 73.7, 79.9, 106.8, 125.4, 125.4, 130.8, 131.7, 133.7, 133.7, 138.5, 164.3, 165.4. ppm. C_14_H_10_N_2_O_2_ (238.24): calcd. C, 70.58; H, 4.23; N, 11.76; found C 70.81; H 4.41; N 11.52.

*(R)-1-(Prop-2-yn-1-yl)-1,10b-dihydropyrimido[2,1-a]isoindole-2,6-dione* [(+)-**8**]: White crystals (51% yield), m.p. 93–95 °C, ${\left[\alpha \right]}_{D}^{20}$ = +368 (c = 0.2, EtOH).

*(S)-1-((1-(2-Methylbenzyl)-1H-1,2,3-triazol-4-yl)methyl)-1,10b-dihydropyrimido[2,1-a]isoindole-2,6-dione* [(−)-**9**]: White crystals (72% yield), m.p. 199–200 °C, ${\left[\alpha \right]}_{D}^{20}$ = −295 (c = 0.2, EtOH), ^1^H-NMR (500 MHz, CDCl_3_, 30 °C): δ = 2.31 (s, 3H, CH_3_), 4.45 (d, *J* = 15.7 Hz, 1H, NCH_2_CH) 5.15 (d, *J* = 15.8 Hz, 1H, NCH_2_CH) 5.43–5.64 (m, 3H, NCH_2_Ar, H-3) 6.18 (s, 1H, H-10b) 7.17–7.31 (m, 4H, Ar) 7.55 (1H, s, CHtriazol) 7.63–7.95 (m, 4H, Ar) 8.70 (d, *J* = 7.9 Hz 1H, H-4), ^13^C NMR (125 MHz, CDCl_3_, 30 °C): δ = 19.1, 37.2, 52.4, 70.3, 106.9, 123.5, 125.0, 126.7, 127.5, 129.2, 129.6, 130.6, 131.1, 131.6, 132.3, 133.6, 133.6, 136.9, 138.9, 143.8, 164.4, 165.8. ppm. C_22_H_19_N_5_O_2_ (385.42): calcd. C, 68.56; H, 4.97; N, 18.17; found C 68.71; H 4.81; N 18.39.

*(R)-1-((1-(2-Methylbenzyl)-1H-1,2,3-triazol-4-yl)methyl)-1,10b-dihydropyrimido[2,1-a]isoindole-2,6-dione* [(+)-**9**]: White crystals (72% yield), m.p. 199–200 °C, ${\left[\alpha \right]}_{D}^{20}$= +287 (c = 0.2, EtOH).

#### 3.2.7. Representative Data for the Racemates (±)-**4**–(±)-**16**


*tert-Butyl ((1R*,2R*,3S*,4S*)-3-(prop-2-yn-1-ylcarbamoyl)bicyclo[2.2.1]hept-5-en-2-yl)carbamate* [(±)-**4**]: Yield 75%, white crystals, m.p. 133–135 °C.

*(1S*,4R*,4aR*,6aS*,12aS*)-6-(Prop-2-yn-1-yl)-1,4,4a,6,6a,12a-hexahydro-1,4-methanoisoindolo[2,1-a]quinazoline-5,11-dione* [(±)-**6**]: Yield 68%, white crystals, m.p. 235–240 °C. 

*(1S*,4R*,4aR*,6aS*,12aS*)-6-((1-(2-Methylbenzyl)-1H-1,2,3-triazol-4-yl)-methyl)-1,4,4a,6,6a,12a-hexahydro-1,4-methanoisoindolo[2,1-a]quinazoline-5,11-dione* [(±)-**7**]: Yield 79%, pale-yellow crystals, m.p. 163–165 °C.

*1-(Prop-2-yn-1-yl)-1,10b-dihydropyrimido[2,1-a]isoindole-2,6-dione* [(±)-**8**]: Yield 68%, white crystals, m.p. 166–168 °C.

*1-((1-(2-Methylbenzyl)-1H-1,2,3-triazol-4-yl)methyl)-1,10b-dihydropyrimido[2,1-a]isoindole-2,6-dione* [(±)-**9**]: Yield 61%, white crystals, m.p. 192–195 °C.

*tert-Butyl ((1R*,2S*,3R*,4S*)-3-(prop-2-yn-1-ylcarbamoyl)bicyclo[2.2.1]hept-5-en-2-yl)carbamate* [(±)-**13**]: Yield 71%, white crystals, m.p. 160–161 °C.

*(1S*,4R*,4aS*,6aR*,12aR*)-6-(Prop-2-yn-1-yl)-1,4,4a,6,6a,12a-hexahydro-1,4-methanoisoindolo[2,1-a]quinazoline-5,11-dione* [(±)-**15**]: Yield 71%, white crystals, m.p. 215–220 °C.

*(1S*,4R*,4aS*,6aR*,12aR*)-6-((1-(2-Methylbenzyl)-1H-1,2,3-triazol-4-yl)-methyl)-1,4,4a,6,6a,12a-hexahydro-1,4-methanoisoindolo[2,1-a]quinazoline-5,11-dione* [(±)-**16**]: Yield 75%, white crystals, m.p. 163–165 °C.

Appendix A contain obtained ^1^H and ^13^C NMR spectra of compounds **4**–**16**, HPLC chromatograms of domino products (+)-**6**, (−)-**6**, (+)-**15** and (−)-**15** and RDA products (+)-**8**, (−)-**8**, (+)-**9** and (−)-**9** presented in the manuscript.

## 4. Conclusions

In summary, we have developed an efficient procedure for the synthesis of novel racemic and enantiomeric 1,2,3-triazole pharmacophore-based pyrimido[2,1-*a*]isoindoles in the reaction of *N*-propargyl-substituted norbornenecarboxamides through a traceless chirality transfer strategy. It involves a three-step reaction using a domino and a copper-catalyzed click reaction as well as an RDA reaction step. The stereochemical information to the newly formed stereogenic centers of the isoindoloquinazolinones was facilitated by the high rigidity of the norbornene skeleton. The absolute configuration of the final products was characterized, because the configuration did not change during subsequent steps. The main advantages of this protocol are simplicity, high yield, short time, mild reaction conditions and easy work-up.

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
