# Peer review of "Synthesis of Novel N-Heterocyclic Compounds Containing 1,2,3-Triazole Ring System via Domino, “Click” and RDA Reactions"

_molecules, 2019, doi:10.3390/molecules24040772_

Round 1
Reviewer 1 Report
The manuscript of Ferenc Fülöp et al. report on the synthesis of novel N-heterocyclic compounds containing 1,2,3-triazole ring system via domino-, "click" and RDA reactions. It is an excellent work, very well designed and carried out, since it considered chiral compounds that maintain their asymmetry even involving domino reactions.
It deserves to be accept for publication, but after some minor changes:
a) Line 39: 1,2,3-disubstituted triazoles… this is not correct, it must be 1,4-Disubstituted 1,2,3-triazoles.
b) Line 112: there is no space between the name and corresponding oxidation state, it should appear as copper(II).
c) Line 130: O,O’- these letters must be italicized.
d) Lines 195 and 205: 28.4, 29.4 and not 28,4 29.4.
e) Line 482: Heterocyclic should be Heterocycl.
f) Line 487: Abbreviation of Asymmetry?
g) It is necessary to numbering the structures that are characterize by NMR to simplify the life of the reader and to be easily identified the assigned proton and carbon-13 signals.
h) In order to improve the quality of the work, the authors could assign the carbon-13 NMR spectra of the synthesized compounds.
Author Response
Answers for Reviewers
Reviewer 1
The manuscript of Ferenc Fülöp et al. report on the synthesis of novel N-heterocyclic compounds containing 1,2,3-triazole ring system via domino-, "click" and RDA reactions. It is an excellent work, very well designed and carried out, since it considered chiral compounds that maintain their asymmetry even involving domino reactions. It deserves to be accept for publication, but after some minor changes:
a) Line 39: 1,2,3-disubstituted triazoles… this is not correct, it must be 1,4-Disubstituted 1,2,3-triazoles.
b) Line 112: there is no space between the name and corresponding oxidation state, it should appear as copper(II).
c) Line 130: O,O’- these letters must be italicized.
d) Lines 195 and 205: 28.4, 29.4 and not 28,4 29.4.
e) Line 482: Heterocyclic should be Heterocycl.
f) Line 487: Abbreviation of Asymmetry?
g) It is necessary to numbering the structures that are characterize by NMR to simplify the life of the reader and to be easily identified the assigned proton and carbon-13 signals.
h) In order to improve the quality of the work, the authors could assign the carbon-13 NMR spectra of the synthesized compounds.
The mistakes (a-f) were corrected, the abbreviation of Asymmetry was checked and it was found to be correct. The numbering of the structures are now included in Scheme 1. The carbon-13 NMR spectra were thoroughly checked. According to publication rules, we would like to give only a list of the carbon-13 signals.
Reviewer 2 Report
Looking for efficient and simple paths for formation of important from different sides molecules plays an important role in contemporary science. The presented material deals with such problem searching for a specific derivatives of N-heterocyclic compounds containing triazole ring. The synthetic approach presented in this manuscript is elegant and opens several further possibilities in utilisation. The whole material looks very solid and the conclusions are sound nevertheless I have not found convincing information about postulated mechanism. It will make a stronger response if the mechanism of the domino reaction (scheme 2) analysis will be deeper. Is it possible to isolate form A? I believe while properly treated this form can be observed especially that the reaction requires rather strong conditions (Scheme 1, v). What about the NMR monitored reaction? In conclusion the presented synthetic approach is nice but the potential impact will be stronger if deeper analysis of performed processes will be presented.
Author Response
Reviewer 2
Looking for efficient and simple paths for formation of important from different sides molecules plays an important role in contemporary science. The presented material deals with such problem searching for a specific derivatives of N-heterocyclic compounds containing triazole ring. The synthetic approach presented in this manuscript is elegant and opens several further possibilities in utilisation. The whole material looks very solid and the conclusions are sound nevertheless I have not found convincing information about postulated mechanism. It will make a stronger response if the mechanism of the domino reaction (scheme 2) analysis will be deeper. Is it possible to isolate form A? I believe while properly treated this form can be observed especially that the reaction requires rather strong conditions (Scheme 1, v). What about the NMR monitored reaction? In conclusion the presented synthetic approach is nice but the potential impact will be stronger if deeper analysis of performed processes will be presented.
The mechanism of the domino reaction is only a proposed reaction pathway. We tried to monitor the 1H NMR reaction as suggested. Unfortunately, we failed to detect the desired intermediates.
A similar reaction pathway was published by Khadem et al. (Tetrahedron Letters 2009, 6661–6664), in which polycyclic compounds bearing the isoindolo[2,1-a]quinoline system were easily prepared in a stereo- and regioselective manner. In the synthetic work 3-ethyl-11-oxo-cyclopenta[c]isoindolo[2,1-a]quinoline carboxylate was synthesized by a multicomponent reaction using ethyl-4-aminobenzoate, 2-carboxybenzaldehyde, and cyclopentadiene as starting materials in acetonitrile, in the presence of trifluoroacetic acid. The authors published the proposed mechanism, in which the first step is the Schiff base formation resulting from the nucleophilic attack of the amine group onto the more reactive carbonyl function of 2-carboxybenzaldehyde, that is onto the aldehyde moiety.
Reviewer 3 Report
The submitted manuscript deals with the synthesis of new derivatives of common heterocyclic systems, in particular 1,2,3-triazoles attached to pyrimidine ring. First of all, I have a big concern on novelty of the chemistry reported here. Moreover, elements of novelty are described in Scheme 1 only; Scheme 3 is redundant because it replicates Scheme 1 for the opposite enantiomer; Scheme 2 reports steps for the domino process, however it was already known as reported in references 45, 47, 48 by the authors of this work, as well as in Eur. JOC, 2018, 4456 (not cited here). From Scheme 1 I can tell that the domino-, “click”-, and RDA-reactions reported are representative examples of well-known chemistries. This is not enough for publishing these results in “Molecules”. The authors should have done at least biological testing of new synthesized derivatives to add value to this work. Secondly, many references are provided for quinazoline heterocycles, but I don’t see any quinazolines in this manuscript (tetracyclic compounds 6 and 7 are not quinazoline heterocycles), which makes a lot of references inappropriate.
In conclusion, I believe the manuscript in its current form is not ready for a publication, at least in “Molecules”.
Author Response
Reviewer 3
The submitted manuscript deals with the synthesis of new derivatives of common heterocyclic systems, in particular 1,2,3-triazoles attached to pyrimidine ring. First of all, I have a big concern on novelty of the chemistry reported here. Moreover, elements of novelty are described in Scheme 1 only; Scheme 3 is redundant because it replicates Scheme 1 for the opposite enantiomer; Scheme 2 reports steps for the domino process, however it was already known as reported in references 45, 47, 48 by the authors of this work, as well as in Eur. JOC, 2018, 4456 (not cited here). From Scheme 1 I can tell that the domino-, “click”-, and RDA-reactions reported are representative examples of well-known chemistries. This is not enough for publishing these results in “Molecules”. The authors should have done at least biological testing of new synthesized derivatives to add value to this work. Secondly, many references are provided for quinazoline heterocycles, but I don’t see any quinazolines in this manuscript (tetracyclic compounds 6 and 7 are not quinazoline heterocycles), which makes a lot of references inappropriate.
In conclusion, I believe the manuscript in its current form is not ready for a publication, at least in “Molecules”.
We cannot agree with the statement: “Scheme 3 is redundant because it replicates Scheme 1 for the opposite enantiomer”. In Scheme 1 the starting derivative is diexo amide (-)-1. The synthesis starting with the opposite enantiomer (+)-1 was also performed, but it was not illustrated in the scheme.
Line 66. “In Schemes 1 and 3 only a single enantiomer is represented for evidence.”
In Scheme 3 the starting derivative was diendo amide (-)-10, which is the diastereomer of diexo amide (-)-1. The domino ring closure products (-)-6 and (-)-15 are diastereomers of methylene-bridged hexahydro isoindolo[2,1-a]quinazolinones, in which 2 chiral centers are the same, whereas 3 are different.
Pentacyclic compounds 6, 7, 15 and 16 are isoindolo-condensed saturated methylene-bridged derivatives of quinazolindione, as can be seen from the names of the ring systems of the compounds.
In the present work our goal was just the synthesis of novel racemic and enantiomeric pyrimido[2,1-a]isoindoles containing the 1,2,3-triazole moiety in the reaction of N-propargyl-substituted norbornenecarboxamides through a traceless chirality transfer strategy. It is our intention to test the biological properties of the compounds in the near future.
The reference Eur. JOC, 2018, 4456 “Nekkaa, I.; Palko, M.; Mandity, I.; Miklos, F.; Fülöp, F. Continuous‐Flow retro‐Diels–Alder Reaction: A Process Window for Designing Heterocyclic Scaffolds. Eur. J. Org. Chem. 2018, 4456–4464.” was added to the manuscript as ref 49.
Round 2
Reviewer 3 Report
We cannot agree with the statement: “Scheme 3 is redundant because it replicates Scheme 1 for the opposite enantiomer”. In Scheme 1 the starting derivative is diexo amide (-)-1. The synthesis starting with the opposite enantiomer (+)-1 was also performed, but it was not illustrated in the scheme.
Line 66. “In Schemes 1 and 3 only a single enantiomer is represented for evidence.”
In Scheme 3 the starting derivative was diendo amide (-)-10, which is the diastereomer of diexo amide (-)-1. The domino ring closure products (-)-6 and (-)-15 are diastereomers of methylene-bridged hexahydro isoindolo[2,1-a]quinazolinones, in which 2 chiral centers are the same, whereas 3 are different.
I can see 1−7, and 10−16 are diastereomers. That was not my question. Schemes 1 and 3 show exactly the same transformations (even with different stereochemistry of compounds), which does not look good and takes unnecessary space in the article. This could have been combined in one scheme or Scheme 3 just mentioned in text (e.g., we have done these transformations for a diastereomer of 1).
My concern about Scheme 2 has not been addressed. It is a reported mechanism, and should not be presented here.
Pentacyclic compounds 6, 7, 15 and 16 are isoindolo-condensed saturated methylene-bridged derivatives of quinazolindione, as can be seen from the names of the ring systems of the compounds.
They can be considered as derivatives of quinazolindione according to nomenclature, but their ring system is not aromatic. If we look into references 1-5, for example, they represent classical aromatic quinazolines, and those compounds have pharmacological importance. Therefore, references 1-5 are inappropriate.
In the present work our goal was just the synthesis of novel racemic and enantiomeric pyrimido[2,1-a]isoindoles containing the 1,2,3-triazole moiety in the reaction of N-propargyl-substituted norbornenecarboxamides through a traceless chirality transfer strategy. It is our intention to test the biological properties of the compounds in the near future.
Why? What is the importance of doing just synthesis of new compounds using well-known chemistry? That was my major concern: lack of novelty. Those kinds of works should not be published in Molecules.
Author Response
Answers rev 3.
We would like to express our thanks for the review and for the critical assessments and the suggestions which will help us improve the quality of our work. Hereinafter, we would like to answer the comments and questions raised by the reviewer.
I can see 1−7, and 10−16 are diastereomers. That was not my question. Schemes 1 and 3 show exactly the same transformations (even with different stereochemistry of compounds), which does not look good and takes unnecessary space in the article. This could have been combined in one scheme or Scheme 3 just mentioned in text (e.g., we have done these transformations for a diastereomer of 1).
In our opinion, when we combined in one Scheme the transformation of the diastereomers of diendo and diexo carboxamides 1 and 5, we may lose the opportunity to show the influence of the different relative configuration of intermediates 6, 7, 15 and 16 to the absolute configuration of the final products 8 and 9. We believe that separate Schemes will help the readers to understand this methodology..
My concern about Scheme 2 has not been addressed. It is a reported mechanism, and should not be presented here.
Scheme 2 was removed as it was suggested.
They can be considered as derivatives of quinazolinedione according to nomenclature, but their ring system is not aromatic. If we look into references 1-5, for example, they represent classical aromatic quinazolines, and those compounds have pharmacological importance. Therefore, references 1-5 are inappropriate.
The references 1-5 were cited upon mentioning some related aromatic quinazolines One sentence and 3 new references were added dealing with saturated quinazolines.
Why? What is the importance of doing just synthesis of new compounds using well-known chemistry? That was my major concern: lack of novelty. Those kinds of works should not be published in Molecules.
To our knowledge, our present synthetic pathway is the first literature example, that contains all three reaction-steps in the same procedure (domino-, click- and RDA-reactions) with a traceless chirality transfer methodology, therefore we feel it is new in literature..